# T-Cell Response against Varicella Zoster Virus in Patients with Multiple Sclerosis during Relapse and Remission

**DOI:** 10.3390/ijms23010298

**Published:** 2021-12-28

**Authors:** Miriam Pérez-Saldívar, Graciela Ordoñez, Benjamín Pineda, Julio Sotelo, Adolfo Martínez-Palomo, José Flores-Rivera, Martha Espinosa-Cantellano

**Affiliations:** 1Department of Infectomics and Molecular Pathogenesis, Center for Research and Advanced Studies (Cinvestav), Mexico City 07360, Mexico; mperezs@cinvestav.mx (M.P.-S.); amartine@cinvestav.mx (A.M.-P.); 2Department of Neuroimmunology, National Institute of Neurology and Neurosurgery “Manuel Velasco Suarez” (INNN), Mexico City 14269, Mexico; gracielaordonez@yahoo.com.mx (G.O.); benpio76@hotmail.com (B.P.); jsotelo@unam.mx (J.S.); 3Clinical Laboratory of Neurodegenerative Diseases, National Institute of Neurology and Neurosurgery “Manuel Velasco Suarez” (INNN), Mexico City 14269, Mexico; jflores.rivera@gmail.com

**Keywords:** virus, autoimmune disease, multiple sclerosis

## Abstract

An association between varicella zoster virus (VZV) and multiple sclerosis (MS) has been reported in Mexican populations. The aim of this study was to compare the response of T cells from MS patients, during relapse and remission, to in vitro stimulation with VZV, adenovirus (AV) and Epstein–Barr virus (EBV). Proliferation and cytokine secretion of T cells from 29 relapsing-remitting MS patients and 38 healthy controls (HC) were analyzed by flow cytometry after stimulating with VZV, AV or EBV. IgG and IgM levels against VZV and EBV were quantified using Enzyme-Linked Immunosorbent Assay. Relapsing MS patients showed a higher percentage of responding CD4+ and CD8+ T cells against VZV compared to AV. In HC and remitting MS patients, proliferation of CD4+ T cells was higher when stimulated with VZV as compared to EBV. Moreover, T cells isolated from remitting patients secreted predominantly Th1 cytokines when cell cultures were stimulated with VZV. Finally, high concentration of anti-VZV IgG was found in sera from patients and controls. The results support previous studies of an VZV-MS association in the particular population studied and provide additional information about the possible role of this virus in the pathogenesis of MS.

## 1. Introduction

Multiple sclerosis (MS) is an immune-mediated demyelinating disease of the central nervous system (CNS) [1]. Although its detailed physiopathology remains unknown, several factors have been reported to be involved in MS, such as failures in central tolerance mechanisms, altered cytokine production and changes in homeostatic proliferation of effector T cells [2,3]. Eighty percent of MS patients have a clinical course characterized by periods of exacerbation followed by remission, termed relapsing-remitting MS (RRMS). Relapses may last for days or weeks and often leave residual disability, which increases as relapses occur [4].

The disease begins with the peripheral activation by unknown stimuli of self-reactive CD4+ T cells, which cross the blood-brain barrier into the CNS [5,6,7]. Once in the CNS, these cells are reactivated by local antigen-presenting cells, resulting in the recruitment of other immune cells, that establish the inflammatory lesion [8,9]. Inflammation is present at all stages of MS, but becomes more evident during acute phases, with the production of high levels of Th1 and Th17 cytokines, including IFN-γ, IL-2 and IL-17 [10,11,12].

Initial activating stimuli have been related to infectious agents [13,14,15], mainly herpes viruses such as varicella zoster virus (VZV) and Epstein–Barr virus (EBV) [16,17,18,19,20].

In our laboratory, we have previously confirmed the presence of VZV in both cerebrospinal fluid (CSF) and peripheral blood mononuclear cells (PBMC), during the first 10 days of an acute relapse in MS patients [21,22]. In contrast, no VZV particles were found in patients with other non-demyelinating neurological diseases that were also under immunosuppressive treatment, excluding non-specific viral reactivation because of immunosuppressive treatments [21]. Moreover, no other herpes viruses (herpes simplex, EBV and human herpes 6 virus) were found in the same cohort of MS patients, either during relapse or remission [21,22,23].

To further analyze the role of VZV in the pathogenesis of MS, we here extend the analysis to evaluate the in vitro proliferative and secretory response of T lymphocyte subpopulations to VZV stimulation in MS patients both during relapse and remission.

## 2. Results

### 2.1. Study Groups

Twenty-nine patients with definite diagnosis of relapse-remitting MS according to McDonald 2010 criteria and 38 healthy controls (HC) were included in the study (Table 1). T cells were stimulated with VZV and the proliferative and secretory responses were compared to stimulation with EBV or AV (only proliferative response). Clinical, demographic and T cell proliferation data of MS patients are described in Appendix A. Higher frequency of proliferative CD4+ and CD8+ T cells against VZV was observed in patients without medical treatment, although without statistical significance (data not shown). A faster increase in Expanded Disability Status Scale (EDSS) was observed in male patients, in accordance with other reports [24].

### 2.2. T Cell Response to Stimulation with VZV

PBMC from 22 MS patients, both during relapse (REL-MS) and remission (REM-MS), 7 relapsing patients and 32 HC, were cultured and stimulated with VZV. There was a higher proliferative response of CD4+ T cells from REM-MS, compared to HC (*p* = 0.0023). For CD8+ T lymphocytes, patients in both relapse and remission showed significantly higher proliferation following VZV stimulation, compared to HC. (Figure 1A,B).

Lower proliferation tended to be registered in Treg cells from REM-MS patients, compared to REL-MS and HC, although this trend did not reach statistical significance (Figure 1C).

### 2.3. T Cell Response to Stimulation with AV

To test the specificity of the response to VZV, PBMC from 13 MS patients in relapse and remission, 6 relapsing patients and 22 HC were also stimulated with adenovirus (AV), a virus unrelated to VZV (Figure 2A–C). Proliferation of CD4+ and Treg cells from MS patients were not significantly different from values obtained from HC. In contrast, CD8+ T cells from REM-MS showed a higher proliferative response than REL-MS and HC.

A paired Wilcoxon test was performed comparing T cell responses to VZV and AV, for each group (REL-MS, REM-MS, and HC). Compared to AV, stimulation with VZV induced higher proliferation in CD4+ (*p* = 0.025) and CD8+ (*p* = 0.012) T cells from REL-MS patients and CD4+ T cells (*p* = 0.049) from REM-MS patients (Figure 2D–F). In contrast, no T cell subpopulation from HC showed significant differences between the response to stimulation with these two viruses (data not shown).

### 2.4. T Cell Response to Stimulation with EBV

PBMC from 9 MS patients both during relapse and remission, 1 relapsing patient, and 10 HC were stimulated with EBV (Figure 3A–C). Compared to HC, CD4+ and Treg cells from REM-MS patients showed a higher proliferative response to stimulation with EBV (*p* = 0.008 and *p* = 0.026, respectively). A paired Wilcoxon test comparing T cell responses to VZV and EBV revealed higher proliferation after stimulation of CD4+ T cells from REM-MS patients and HC with VZV (*p* = 0.039 and *p* = 0.016, respectively) (Figure 3D,E). No difference was observed after stimulation of CD8+ T cells for either group (REL-MS, REM-MS, and HC) (data not shown).

With the data obtained from all proliferations, we ran paired Wilcoxon tests to compare the response of relapse and remission cells from each subject (CD4+, CD8+ and Treg cells) to stimulation with each virus (VZV, AV and EBV). Higher proliferative responses tended to be observed in REM-MS patients, although statistical significance was only reached in CD8+ T cells after AV stimulation (Appendix A).

### 2.5. Cytokine Cuantification in Stimulated Culture Cells

We quantified IL-17, IFN-γ, TNF, IL-10, IL-4, IL-6, and IL-2 in cell culture supernatants of PBMC from patients and controls stimulated either with VZV or EBV. Only IFN-γ, TNF and IL-2 were detected in supernatants of PBMC from REM-MS patients stimulated with VZV (Figure 4A). On the other hand, non-significant concentrations of TNF, IL10 and IL-2 were detected after EBV stimulation (Figure 4B). Very low levels of cytokines were obtained in cell cultures of HC and REL-MS.

IL-6 was secreted at high concentrations in both patients and controls, without statistical differences after stimulation with VZV or EBV (data not shown).

### 2.6. Specific IgG and IgM Serum Antibodies against VZV and EBV

Antibodies against VZV and EBV were measured in the sera of all individuals. We found no difference between HC, REL-MS, and REM-MS for anti-VZV IgM and IgG (Figure 5A,B). In contrast, a higher concentration of IgM against EBV was found in REL-MS compared to HC. This difference decreased in patients during remission (Figure 5C), although this concentration does not reach the lower limit to consider an active infection with this virus.

Interestingly, 95% of the HC were positive for anti-VZV IgG. However, only 82% of them were positive for IgG against EBV. Except for one patient, none of the patients or controls had received the VZV vaccine at the time the blood sample was taken. Therefore, the immunological memory found from the quantification of anti-VZV IgG reflects a previous natural infection.

## 3. Discussion

In this study we found high proliferative responses of CD4+ and CD8+ T cells in MS patients after VZV stimulation in comparison to HC, suggesting reactivation of this virus in these patients. Considering that VZV establishes latency in dorsal root ganglia, an increase of memory T cells against VZV could be present in MS [25].

There was a predominance of CD8+ T cell response upon stimulation with each of the three viruses (VZV, AV and EBV). These cells recognize MHC class I-peptide complexes on the surface of any infected cell, not only antigen-presenting cells, which might contribute for a greater CD8+ T cell expansion [26].

The lower percentage of proliferating T cells after viral stimulation observed in REL-MS compared to REM-MS may reflect either the selective migration of T cells to the CNS [27] or a T cell exhaustion caused by persistent viral infection [28,29]. A proliferative anergy of circulating T cells in remitting MS patients, and their reactivation by an agonistic anti-CD28 antibody has been reported [30]. High proliferation of effector T cells from MS patients during remission in response to viral stimulation, mainly VZV, supports the theory of an extrinsic factor that induces the change from remission to relapse [31,32].

With respect to proliferation assays, the difference between stimulation with VZV or EBV was not as evident as between VZV or AV for MS patients. However, serology results suggest a greater immunological memory against VZV compared to EBV in this particular Mexican population. A total of 108,602 cases of VZV were reported in Mexico in the 2019 Epidemiological Bulletin [33], and although there are no exact data on the seroprevalence of EBV, it is also a widely distributed virus in the country.

Remarkably, in HC we found a lower percentage of proliferating CD4+ T cells in cultures stimulated with EBV compared to VZV. This correlates with the serological assay since there was a lower frequency of HC positive to anti-EBV IgG compared to VZV.

The cytokine quantification in culture supernatants of PBMC of patients and controls stimulated with VZV confirm the results obtained from proliferation assays. A Th1 cytokine profile was predominant in cell cultures of REM-MS patients stimulated with VZV, in agreement with studies of systemic memory T-cells expressing IFN-γ, TNF and IL-2 in latent VZV infected individuals [34] and coinciding with reports that MS should be considered a Th1/Th17 driven disease [35,36]. Hence, VZV could be driving or enhancing Th1 inflammation, possibly aided by cytokines secreted by B cells or macrophages, since our cultures contained not only T lymphocytes, but all PBMC.

High production of TNF-α in REM-MS cell cultures after VZV stimulation could be related to a significant increase in TNF mRNA expression in PBMC of 29 MS patients, prior to a relapse [37], suggesting the participation of VZV in both the development of MS and its course. The altered immune response could also be related to a molecular mimicry mechanism, which has been reported in MS [38,39,40].

Although proliferation of effector T cells from control subjects was higher when stimulated with VZV compared to EBV, cytokine secretion in these cultures was low. This might suggest the participation of an unknown factor that favors the production of pro-inflammatory cytokines in the presence of VZV in MS patients.

A failure in the suppressive function has been described in CD4 + CD25 + FoxP3+ T cells from MS patients, mainly with a deficient synthesis of immunosuppressive cytokines (IL-10, IL-35, and TGF-β) [41,42]. However, in this study the proliferation percentages of Treg cells and IL-10 production were comparable between MS patients and HC.

Our results are consistent with the immunopathology of MS described so far, since peripheral CD4+ T cells have been identified as responsible for initiating the inflammatory process in MS [8], but the lymphocytic infiltrates in MS patients’ lesions are constituted mainly by CD8+ T cells [43,44], which showed high proliferation in the present study. Moreover, the presence of shared CD8+ T cell clones between peripheral blood and CNS has been demonstrated [45].

Altogether, these results support previous studies of a VZV-MS association and provide additional information about the role of this virus in the pathogenesis of MS.

## 4. Materials and Methods

### 4.1. Patients and Control Subjects

Twenty-nine patients with definite diagnosis of relapse-remitting MS according to McDonald 2010 criteria were studied within the first 10 days of an acute relapse (REL-MS) before receiving steroid treatment. A second blood sample was obtained from 22 of them at least one month after, while they were on remission (REM-MS). Clinical, demographic and T cell proliferation data of MS patients are detailed in Appendix A. Thirty-eight healthy subjects within the same age range of MS patients and same female/male ratio were included as controls (HC). All subjects were Mexican and of Mexican ascent at least up to grandparents. The study was carried out following the rules of the Declaration of Helsinki and was approved by the Ethics Committee of the National Institute of Neurology and Neurosurgery. Written informed consent was obtained from all subjects.

The main objective of this study was to evaluate the response of T cells from MS patients to VZV stimulation, given the association of this virus with MS in the Mexican population. We initially compared the response to VZV stimulation with AV, since this virus, as with other respiratory viruses, has been associated with an increased risk of relapse [31]). After testing a first group of patients, we put together a second group to compare VZV with EBV, a virus also associated with MS and belonging to the same family of herpesvirus.

### 4.2. Blood Samples

Peripheral blood samples were obtained by venipuncture. For each subject, two tubes of four milliliters were extracted, with and without anticoagulant. After blood clot formation, the second tube was centrifuged for 3 min at 3000× *g* to obtain serum, which was stored at −80 °C until use.

### 4.3. Isolation of PBMC

PBMC were isolated by density gradient centrifugation on Ficoll, then resuspended in Recovery Cell Culture Freezing medium (Thermo Fisher Scientific, Waltham, MA, USA), and stored at −80 °C for further analysis.

### 4.4. Total Proteins Quantification in Virus Sources

Quantification of total proteins in VZV vaccine (Oka) Varivax^®^ III (MSD, Kenilworth, NJ, USA), AV suspension (ATCC^®^ VR-1516™) and EBV suspension (ATCC^®^ VR-603™) was determined by triplicate using the Qubit 3.0 fluorometer. Average concentrations were as follows: VZV vaccine 4.3 µg/µL, AV suspension 1.32 µg/µL and EBV suspension 9.4 µg/µL.

### 4.5. Proliferation Assays

PBMC from the Cell Culture Freezing medium were thawed, washed with RPMI 1640 supplemented with 10% fetal bovine serum and 1% antibiotic (complete medium), and counted using trypan blue. For each patient (relapse and remission) and control, 16 wells of a 96-well round-bottom culture plate were seeded with 1 × 10^5^ PBMC in 200 µL of complete medium. Plates were incubated 24 h at 37 °C in a 5% CO_2_ humidified atmosphere, for total cell recovery. After incubation, four conditions were analyzed: (1) virus-free medium (negative control), (2) Concanavalin A (Sigma-Aldrich, St. Louis, MO, USA) at a concentration of 10 µg/mL (positive control), (3) VZV vaccine (volume equivalent to 1.25 µg of total protein), and (4) 1.25 µg of total protein from either AV or EBV. Four wells were seeded for each condition (two for CD4+ and CD8+ T cells determination and the other two for regulatory T cells). Plates were incubated at 37 °C in a 5% CO_2_ humidified atmosphere for 5 days. At the third day of incubation, 10 µM of the thymidine analog BrdU (BD, Franklin Lakes, NJ, USA) was added to each well to evaluate T cell proliferation.

### 4.6. Flow Cytometry

After 5-day incubation, cells were stained with anti-CD8 PerCP-Cy5.5, CD4 PE, CD25 APC and anti-BrdU FITC (BD, Franklin Lakes, NJ, USA); while regulatory T cells (Treg) were identified with anti-CD4 PE, FoxP3 PerCP-Cy5.5, CD25 APC (BD, Franklin Lakes, NJ, USA). Samples were analyzed by flow cytometry. 10,000 events were registered within the lymphocyte region identified according to their size (Forward-scatter, FCS) and complexity (Side-scatter, SSC), considering their activation status [46].

Data were analyzed with FlowJo software (v.10; Tree Star, Inc., Ashland, OR, USA).

### 4.7. Quantification of Cytokine Secretion

Four wells of a 96-well round-bottom culture plate were seeded with 1 × 10^5^ PBMC in 200 μL of complete medium, stimulated as in the proliferation assays (negative control, positive control, VZV or EBV), and incubated for 48 h at 37 °C in a humid atmosphere and 5% CO_2_. Cell cultures were harvested and centrifuged at 10,000× *g* for 3 min. Supernatants were collected for quantification of IL-2, IFN-γ, TNF-α, IL-17, IL-10, IL-4, and IL-6, using a CBA Human Th1/Th2/Th17 Cytokine Kit (BD, Franklin Lakes, NJ, USA) according to the manufacturer’s instructions.

### 4.8. IgG and IgM Anti-VZV and EBV

IgG and IgM antibodies against VZV and EBV were determined in sera from patients and controls using commercial enzyme-linked immunosorbent assay (ELISA) kits (IBL International GMBH, Hamburg, Germany; detection range: 1–150 U/mL). Concentration in U/mL was obtained using the Four Parameter Logistic (4PL) regression model (AAT Bioquest, Inc., Sunnyvale, CA, USA).

### 4.9. Statistical Analysis

Statistical analysis was performed with GraphPad Prism 6.01 software. Due to the age range in the study group, we evaluated whether this covariant could influence our results using linear regression analysis on MS patients and controls. There was no correlation between the age of patients or controls and the proliferative response or anti-VZV or -EBV antibody concentration.

Differences in proliferation and IgG and IgM concentration between two groups were evaluated using the non-parametric Mann-Whitney ranked test. Paired samples (VZV-AV, VZV-EBV and REL-REM MS) were tested with the non-parametric Wilcoxon ranked test. Significance was considered with *p* < 0.05.

The response of T cell subpopulations to stimulation with Concanavalin A confirmed the ability of cells to proliferate (positive control) but was not included in the statistical analysis.

Cytokine concentrations were compared with two-way ANOVA tests, using Bonferroni correction for multiple comparisons. A value of *p* < 0.05 was considered significant.

Values obtained in unstimulated cells were subtracted to those obtained in the cultures stimulated with VZV, AV and EBV.

## Figures and Tables

**Figure 1 ijms-23-00298-f001:**
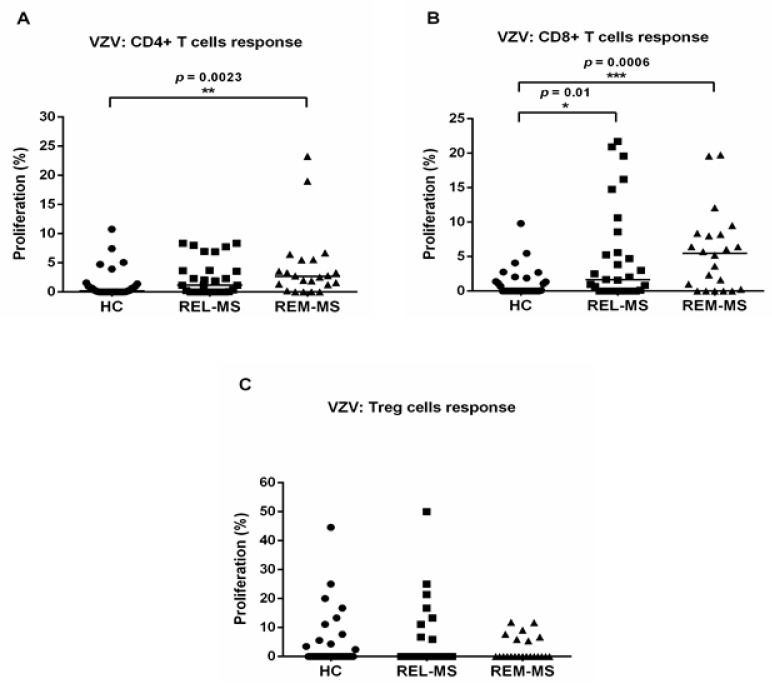
T cell response of REL-MS (*n* = 29), REM-MS (*n* = 22) and HC (*n* = 32) to stimulation with VZV. Proliferation of CD4+ T cells was significantly higher in REM-MS patients compared to HC (**A**). CD8+ T cells from MS patients (REL-MS and REM-MS) showed a higher proliferative response compared to HC (**B**) and there was no significant difference in Treg cell response between the three groups (**C**). Data are presented as the mean of proliferation percentage. Each dot represents one subject, and horizontal bars correspond to the median values. * *p* < 0.05, ** *p* < 0.01, *** *p* < 0.001.

**Figure 2 ijms-23-00298-f002:**
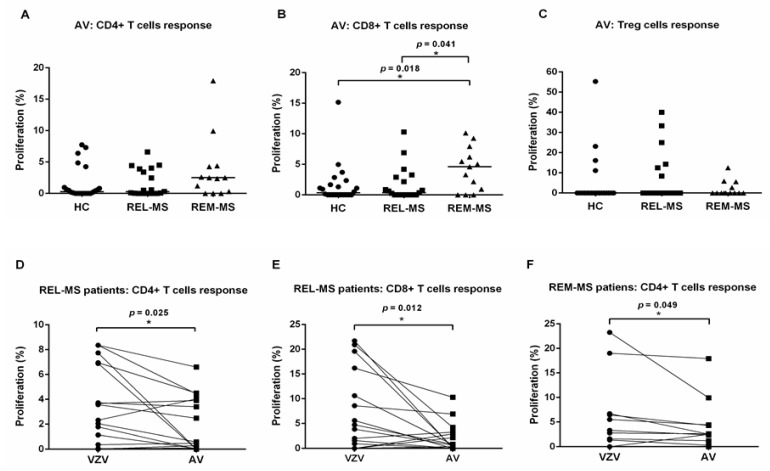
T cell response of REL-MS (*n* = 19), REM-MS (*n* = 13) and HC (*n* = 22) to stimulation with VZV or AV. Proliferation of CD4+ and Treg cells was comparable among all groups (**A**,**C**). In contrast, CD8+ T cells from REM-MS patients showed higher proliferation after AV stimulation, compared to REL-MS and HC (**B**, *p* < 0.04 and *p* < 0.02, respectively). Horizontal bars correspond to the median values. Paired Wilcoxon tests comparing T cell responses to VZV and AV, for each group (REL-MS, REM-MS, and HC) revealed that effector CD4+ and CD8+ T cells from REL-MS, and CD4+ T cells from REM-MS patients showed higher proliferation in response to stimulation with VZV (**D**–**F**). Each dot represents one subject. * *p* < 0.05.

**Figure 3 ijms-23-00298-f003:**
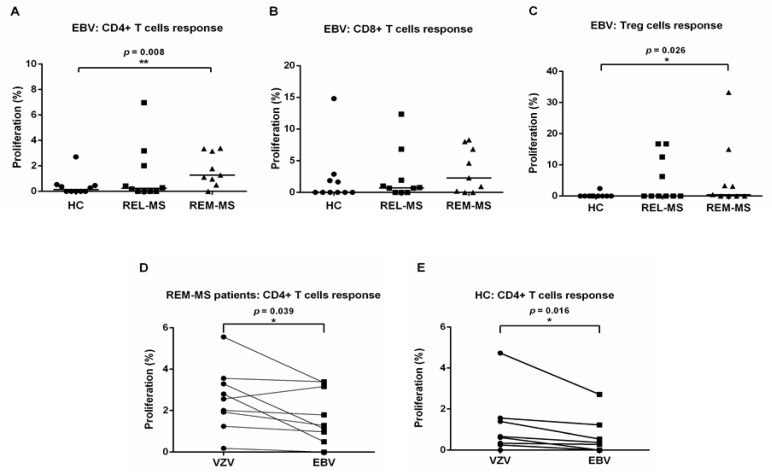
T cell response of REL-MS (*n* = 10), REM-MS (*n* = 9) and HC (*n* = 10) to stimulation with EBV. CD4+ and Treg cells from REM-MS patients showed a higher proliferative response to stimulation with EBV compared to HC (**A**,**C**), while no difference was observed after stimulation of CD8+ cells (**B**), or when testing cells of REL-MS patients. A paired Wilcoxon test comparing T cell responses to VZV and EBV stimulation for all groups (REL-MS, REM-MS, and HC) revealed that only CD4+ T cells from REM-MS patients and HC showed a statiscally higher proliferation after stimulation with VZV (**D**,**E**). No differences were observed in any other group (REL-MS, CD8+ or Treg from REM-MS and HC (data not shown). Data are presented as the mean of proliferation percentage. Each dot represents one subject. * *p* < 0.05, ** *p* < 0.01.

**Figure 4 ijms-23-00298-f004:**
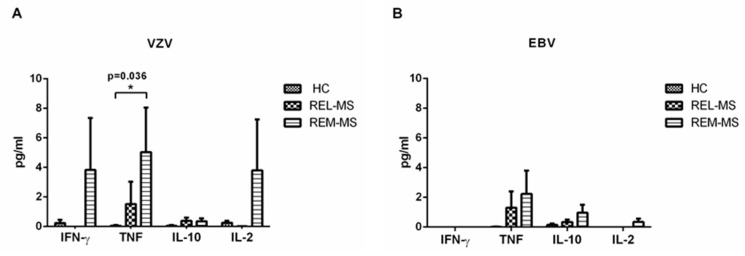
Cytokines secreted in PBMC cultures stimulated with VZV or EBV. A predominance of IFN-γ, TNF and IL-2 was observed in cell cultures of REM-MS patients stimulated with VZV (**A**). Low concentration of cytokines was obtained in cultures stimulated with EBV (**B**). * *p* < 0.05.

**Figure 5 ijms-23-00298-f005:**
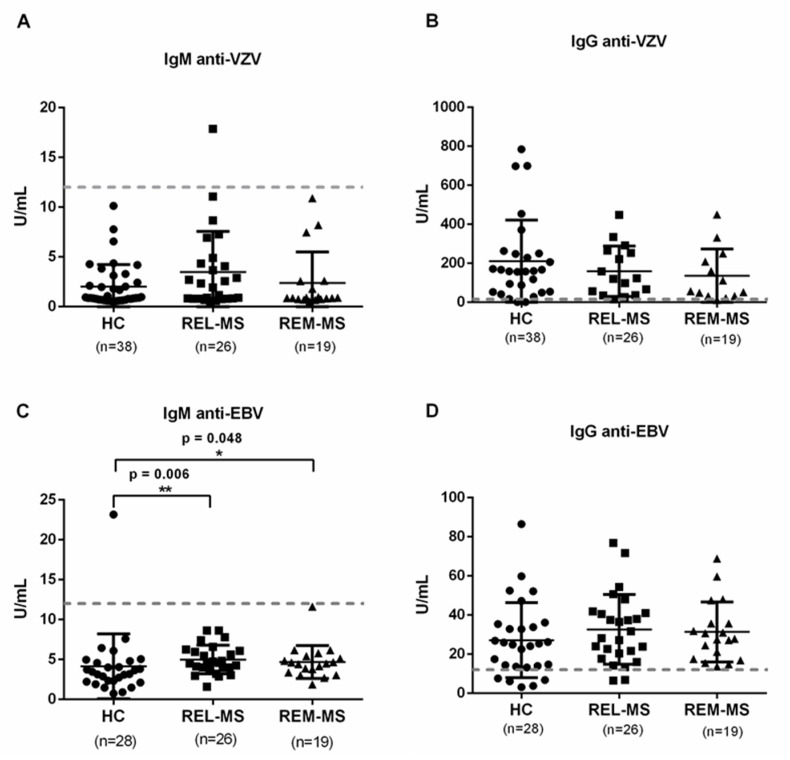
Serum IgM and IgG. Sera from REL-MS, REM-MS patients and HC were collected to quantify antibodies against VZV (**A**,**B**) and EBV (**C**,**D**) by Enzyme-linked immunosorbent assay. MS patients had higher concentration of anti-EBV IgM in comparison with HC. Dotted lines indicate the concentration above which a sample is considered positive, according to the manufacturer’s instructions. * *p* < 0.05, ** *p* < 0.01.

**Table 1 ijms-23-00298-t001:** Clinical and demographic data of MS patients and HC.

Study Groups	Age ^a^	Gender (F/M)	Evolution (Years) ^a^	Relapses/Year ^a^	EDSS ^a^	T/NT
MS (*n* = 29)	30 ± 7.3	18/11	5.8 ± 5.8	1.6 ± 0.9	3.4 ± 2.1	15/14
HC (*n* = 38)	27.1 ± 4.8	23/15	NA	NA	NA	NA

^a^ Data shown as the mean ± standard deviation. EDSS: expanded disability status scale T: treated. NT: non-treated. NA: Not applicable.

## Data Availability

Publicly available datasets were analyzed in this study. This data can be found on the following link: https://cinvestav365-my.sharepoint.com/personal/mespinosac_cinvestav_mx/_layouts/15/onedrive.aspx?id=%2Fpersonal%2Fmespinosac%5Fcinvestav%5Fmx%2FDocuments%2Fijms%2D1418986%20Raw%20data.

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
