# Peer review of "T-Cell Response against Varicella Zoster Virus in Patients with Multiple Sclerosis during Relapse and Remission"

_ijms, 2021, doi:10.3390/ijms23010298_

Round 1

Reviewer 1 Report

The manuscript by Pérez-Saldívar et al. describing how circulating T cells from MS patients are activated and secrete Th1 cytokines in response to varicella zoster virus than against adenovirus or Epstein-Barr virus is well written and interesting, as viral infections are known risk factors for MS.

I would only ask the authors to please, explain how the healthy controls were selected (were they sex- and age-matched to the patients?).

I would also say that I could not open some of the files included as Supplementary Material, and that I have noticed that Supplementary Figures 1 and 2 are not mentioned in the text of the manuscript.

Author Response

The manuscript by Pérez-Saldívar et al. describing how circulating T cells from MS patients are activated and secrete Th1 cytokines in response to varicella zoster virus than against adenovirus or Epstein-Barr virus is well written and interesting, as viral infections are known risk factors for MS.

We thank reviewer 1 for his/her comment.

I would only ask the authors to please, explain how the healthy controls were selected (were they sex- and age-matched to the patients?).

Yes. Healthy controls were sex- and age- matched to patients. This is now stated in Materials and Methods, while describing the control group.

I would also say that I could not open some of the files included as Supplementary Material, and that I have noticed that Supplementary Figures 1 and 2 are not mentioned in the text of the manuscript.

We apologize for this. We are resending all Supplementary Material, which is now mentioned in the text.

Reviewer 2 Report

In this manuscript Perez-Saldivar et al compared the response of T cells, both CD4 and CD8, isolated from MS patients and healthy control to VZV, AV and EBV. Results are difficult to understand in the present form. The major issue of the manuscript is that it is not clear why the authors divided the samples of MS patients into to cohort and why the response of PBMCs from MS to VZV was analyzed maintaining the two cohorts separate. Looking at the proliferative response to VZV they obtained contrasting results between the two cohorts, but no explanation on this is present in the discussion section.

Specific points:

1) Line 61-63: It is not clear to me how patients were assigned to the two cohorts. Authors state that VZV and AV were used to stimulate cells from the first cohort of patients and VZV and EBV for the second one. Why? In other words why the impact of VZV, AV and EBV were not analyzed at the same time in all REL-MS, REM-MS and HC? This should be clarified in the text. At present the division of patients into two cohorts make no sense.

To compare the effects of VZV, EBV and AV it is fundamental to test them in the same patients, since individual characteristics may affect the results.

2) Both in Fig 1A and Fig 3A authors assess the response of CD4+T cells from MS and HC to VZV, and obtained opposite results. Why not analyze the response of CD4+T from all patients (REL-MS=29 REM-MS=22) to VZV in a same graph? The same consideration is valid for Fig 1C and 3C where they show CD8+T cells responses. Authors should add a comment on the opposite response of MS cells from cohort A and cohort B to VZV.

3) Line 84: Please modify “proliferative percentage of of CD4+T cells” I should say “Proliferation of CD4+ T cells was comparable...”

Author Response

Response to Reviewer 2 Comments (also uploaded as an attachment)

The major issue of the manuscript is that it is not clear why the authors divided the samples of MS patients into to cohort and why the response of PBMCs from MS to VZV was analyzed maintaining the two cohorts separate. Looking at the proliferative response to VZV they obtained contrasting results between the two cohorts, but no explanation on this is present in the discussion section.

The reviewer is right, and we should have thought of combining the two groups in the first place. The reason to present the two cohorts was solely a reflection of the chronological development of our experiments. We originally concentrated on the analysis of T cell proliferative response to stimulation with VZV. A group of 19 patients was gathered and an unrelated virus (AV) was used as a control. During analysis, we thought it would be very interesting to study the proliferative response to stimulation with EBV. Being from the same family as VZV (herpes virus) and having been associated to MS in other populations, we thought it was worth extending the analysis. We had run out of cells from this group and it took longer to gather the second group of only 10 patients. By then, resources were scarce, PBMC cells were used in another MS project, and only EBV stimulation could be included.

By combining both cohorts, subtle differences mentioned in the original manuscript as a tendency without statistical significance (higher frequency of proliferative CD4 + and CD8 + T cells against VZV or faster increase EDSS in male patients from one vs the other cohort) were no longer maintained, indicating that these differences were indeed not significant.

Specific points:

1) Line 61-63: It is not clear to me how patients were assigned to the two cohorts. Authors state that VZV and AV were used to stimulate cells from the first cohort of patients and VZV and EBV for the second one. Why? In other words why the impact of VZV, AV and EBV were not analyzed at the same time in all REL-MS, REM-MS and HC? This should be clarified in the text. At present the division of patients into two cohorts make no sense.

To compare the effects of VZV, EBV and AV it is fundamental to test them in the same patients, since individual characteristics may affect the results.

Specific point 1: As mentioned above, the two cohorts reflected the chronological development of the experiments. However, we have now grouped all patients together. It would have been better to compare the effects of VZV, EBV and AV in the same patients, but, as explained, patient´s PBMC cells were also used in another MS project, and only EBV stimulation could be included in the second group of patients.

2) Both in Fig 1A and Fig 3A authors assess the response of CD4+T cells from MS and HC to VZV, and obtained opposite results. Why not analyze the response of CD4+T from all patients (REL-MS=29 REM-MS=22) to VZV in a same graph? The same consideration is valid for Fig 1C and 3C where they show CD8+T cells responses. Authors should add a comment on the opposite response of MS cells from cohort A and cohort B to VZV.

Specific point 2: The idea of reviewer 2 to combine both cohorts was very much appreciated. Following his/her suggestion to analyze the response of CD4+, CD8+ and Tregs from all patients (REL-MS=29 REM-MS=22) to VZV in a same graph, we restructured the analysis to present proliferative results after stimulation with VZV, EBV or AV each in one graph. Results are now much clearer to interpret.

By analyzing all patients together, subtle differences mentioned in the original manuscript as a tendency without statistical significance (higher frequency of proliferative CD4 + and CD8 + T cells against VZV) were no longer maintained.

3) Line 84: Please modify “proliferative percentage of of CD4+T cells” I should say “Proliferation of CD4+ T cells was comparable...”

Specific point 3: Text has been changed.

Round 2

Reviewer 2 Report

The manuscript has been improved and it is acceptable for publication